# Experiencing Food Restrictions for Health and Weight Control in Childhood and Their Links to Restrained Eating and Excessive Body Weight in Polish Young Adults—A Cross-Sectional Study

**DOI:** 10.3390/nu17010087

**Published:** 2024-12-29

**Authors:** Marzena Jeżewska-Zychowicz, Aleksandra Małachowska, Marta Sajdakowska, Zuzanna Żybula

**Affiliations:** 1Institute of Human Nutrition Sciences, Warsaw University of Life Sciences (SGGW-WULS), Nowoursynowska 159 C, 02-776 Warsaw, Poland; marzena_jezewska_zychowicz@sggw.edu.pl (M.J.-Z.); s205460@sggw.edu.pl (Z.Ż.); 2Military Institute of Medicine—National Research Institute, Zegrzyńska 8, 05-119 Legionowo, Poland; ola.malach@gmail.com

**Keywords:** feeding-related behaviors, body weight, overweight, youth, dietary restraint

## Abstract

Objectives: A better understanding of the conditions leading to overweight and obesity is fundamental due to the ever-increasing phenomenon of excess body weight. This study aimed to determine how the occurrence of restrained eating in young adulthood, food-restricted types, and food experiences related to food restrictions originating in childhood correlate with excessive body weight among young adults. Methods: The data were collected in Poland in 2020–2021 using a Computer-Assisted Web Interview technique as a cross-sectional study among 358 young adults. Descriptive statistics, frequency analysis, cross-tabulations, and logistic regression analysis were performed. Results: Restrictions associated with limiting intake of sugar-rich foods were more characteristic of those with normal body weight (52.7%) than those with excessive body weight (39.2%). Weak positive correlations were found between childhood experiences of weight control restrictions and restrained eating (0.222), using food restrictions (0.143), the number of restrictions (0.152), using restrictions on sugar (0.149), and food rich in fat (0.105) in early adulthood. Childhood food experiences related to weight control restrictions favored having excessive body weight in young adults by 53% after adjusting for sex and age. Conclusions: The results showed that it is advisable to develop interventions to increase parents’ awareness of the possible long-term consequences of practices related to controlling children’s eating behavior.

## 1. Introduction

The increase in the prevalence of overweight and obesity in most countries has been a cause of great concern globally [1,2]. Obesity is a significant public health problem that results in decreased life expectancy at all ages, for both women and men, in minority populations, and also in the poor as compared to the prosperous [3,4]. Obesity is linked to the occurrence of chronic medical problems, impairment of health-related quality of life, and increased healthcare and medication spending [5,6,7]. Obesity is today’s most significant public health challenge due to the rapid increase in children and young adults with obesity [8]. In addition, children with obesity tend to remain obese in adulthood and are at increased risk of diabetes and cardiovascular problems at a younger age [9]. 

Over the last few decades, the environment has changed substantially in terms of easy access and affordability of high-calorie fast food and ultra-processed food, increased portion sizes, increased intake of sugary drinks, and sedentary lifestyles, which contribute to an increased prevalence of obesity [10]. Consumption of high-calorie, energy-dense food options, and reduced physical activity may contribute to a sustained positive energy balance. However, limited access to inexpensive and high-quality food (e.g., supermarkets), referred to as a ‘food desert’ [11], is also positively associated with obesity. In addition to food availability, the type of food is also relevant to the obesity epidemic. For example, a higher risk of weight gain is associated with increased consumption of sugar-sweetened beverages [12], high-sugar food [13], sweets [14], ultra-processed food [15,16], or fried food [17]. The widespread availability of weight-promoting foods is one element of the obesogenic environment and should, therefore, be a strong focus of strategies to combat the obesity epidemic. 

As major food providers, models, and regulators regarding food intake, parents influence children’s eating in various ways [18,19,20]. Parental feeding practices are a source of childhood food experiences, which can influence eating behavior in youth and later adulthood. As gatekeepers, parents can limit their child’s access to some food through restriction practices. However, it is still unclear how restrictions are linked to children’s adequate food intake, especially when long-term effects are considered [21]. Parental content-restricted feeding may increase the attractiveness and intake of foods that have been previously restricted [22]. 

Restrictions, however, may not lead to the expected outcomes. Restrictions, especially those related to weight control, may contribute to less reliance on internal cues and eating for reasons unrelated to physical hunger [23,24]. It may lead to overeating, which is described as a restraint theory [25]. The use of restrictions to control weight is a part of restrained eating. Restrained eating is an eating style that describes psychological aspects of dietary behaviors, such as beliefs, attitudes, motives, and feelings toward food and eating [26]. Although eating restraint reflects individuals’ cognitive and behavioral efforts to control their weight [27], it does not always lead to a lower BMI [28]. The impact of restrained eating on food intake was confirmed in previous research [29,30,31]. Restrained eating in adult women and men may be related to their recollection of parents using food to control their childhood behavior by restricting certain food intake [32,33]. 

Results of previous studies suggest that experiencing food restriction in childhood can result in restricting food intake in adulthood, especially foods that negatively affect body weight. As restrained eating often refers to high-energy foods, consideration of this concept alongside childhood experiences may help explain the prevalence of excessive body weight. Thus, the study aimed to assess how restrained eating in young adulthood, food-restricted types, and food experiences related to food restrictions originating in childhood are linked with excessive body weight among young adults.

## 2. Materials and Methods

### 2.1. Study Design and Participants

The data were collected in Poland in 2020–2021 using a Computer-Assisted Web Interview technique as a cross-sectional study. An invitation to participate in the survey with a short description of its purpose and a link to a questionnaire was published on social media. The inclusion criteria were people over 18 and below 25, women and men, and informed consent being provided for participation in the study. The exclusion criteria were as follows: age below 18 and over 25, no consent for participation in the study. Data anonymity and confidentiality were assured. The sample size was calculated to provide 95% power to detect a medium effect size (w = 0.3) using chi-squared tests (df = 4, significance level a = 0.05). The minimum sample size was 207. The invitation to the study accepted 435 respondents who correctly filled out the questionnaire. From this group, 77 respondents were excluded from the sample due to their inability to remember food experiences from childhood related to restrictions, i.e., those who answered “I don’t remember” for any question or statement related to food experiences. The final study sample included 358 young adults who declared they had experienced health and weight control restrictions in childhood. The study design was approved by the Ethics Committee of the Institute of Human Nutrition Sciences, Warsaw University of Life Sciences, in Poland (Resolution No. 02/2020). It was conducted in compliance with the Declaration of Helsinki.

### 2.2. Questionnaire

The questionnaire included questions on experiences of food restrictions in childhood and early adulthood. Experiences of food restrictions in childhood were measured based on parental feeding practices related to food restrictions remembered from childhood. The questionnaire was developed using the Comprehensive Feeding Practices Questionnaire (CFPQ) [34] and validated in Polish adults [35]. Respondents’ childhood food restrictions experiences were measured with 12 statements (Appendix A: Questionnaire). Respondents were asked to agree/disagree with the sentences describing family habits from the period of their childhood using a 6-point scale: 1—“disagree”; 2—“slightly disagree”; 3—“neither agree nor disagree”; 4—“slightly agree”; 5—agree”; 6—“I don’t remember”. The answer “I don’t remember” was used as an exclusion criterion when the analysis was carried out. 

Experiences of food restrictions in early adulthood were tested using restrained eating and four questions about restricting four food groups. Restrained eating was assessed with the validated Polish version of the Dutch Eating Behavior Questionnaire [29,36]. The scale contains 10 items rated with a 5-point Likert scale, ranging from never (1) to very often (5). Behaviors related to the restrictions on eating four food groups, i.e., processed foods, sugar, sugar-rich foods, and fat-rich foods, were assessed based on the question “What restrictions do you apply to your food intake?” with the following responses available: I restrict; I do not apply the restriction. Based on the respondents’ answers, the following variables were developed: restrictions on eating foods (at least one restricted food/no foods restricted); restrictions on eating particular foods, i.e., processed foods, sugar, sugar-rich foods, fat-rich foods (Yes/No); and the number of restrictions applied (range 0–4 restricted foods).

In addition, the questionnaire included questions about respondents’ socio-demographic characteristics, including sex, age, residence, and education. Respondents reported their weight and height. Body mass index (BMI) was calculated and interpreted according to the criteria of the World Health Organization [37]. A BMI of 18.5–24.9 kg/m^2^ was assessed as normal weight, while a BMI of 25.0 kg/m^2^ and above was considered excessive weight.

### 2.3. Statistical Analysis

Questions and statements about childhood food experiences were assigned to two scales, i.e., Restrictions for Health and Restrictions for Weight Control [34,35]. Scores for “Restriction for health” and “Restriction for weight control” were calculated by summing the individual scores for each respondent, and then mean values were calculated. The scores ranged from 1 to 5. The higher the score, the more experiences resulted from specific parental feeding practices during childhood. Scores for Restrained eating were calculated by summing the individual scores for all statements from the scale, and then the mean value was calculated. The resulting scores ranged from 1–5. Cronbach’s alpha reliability coefficients were calculated for each of the scales. The degree of fit of the scales was satisfactory (Cronbach’s alpha: 0.709, 0.833, and 0.899, respectively). 

Descriptive statistics, frequency analysis, and cross-tabulations were performed. The Shapiro–Wilk test was used to test the normality of distribution. A chi-squared test was used, with an accepted significance level of *p* < 0.05. The Spearman correlation coefficient was used to assess the bivariate correlations between variables.

Logistic regression analysis was used to verify the association between excessive body weight (dependent variable) and the following exploratory (independent) variables: restrained eating, childhood experiences of restrictions for weight control and health, restrictions on eating foods, and several restrictions applied. Odds ratios (OR) represented the probability of belonging to a group with excessive body weight. The reference groups (OR = 1.00) were those of normal weight and did not use restrictions. The continuous variables were restrained eating, childhood experiences of restrictions for weight control and health, and several restrictions applied. Both crude and adjusted (for sex and age) were developed. Only statistically significant variables at the significance level of α = 0.05 were included in the models. Wald’s test was used to assess the significance of ORs. The Hosmer–Lemeshow goodness-of-fit test, the explained variation of survival (Nagelkerke R2), and the correctness of predictions were used to assess the quality of the resulting model [38].

Statistical analysis was conducted using IBM SPSS Statistics for Windows, version 29.0 (IBM Corp, Armonk, NY, USA).

## 3. Results

### 3.1. Description of the Study Sample

Table 1 presents the characteristics of the study sample. The sample consisted of 358 adults aged 18–25 years old, with 50.8% of female respondents. The median age was 21, while the median BMI was 22.6 kg/m^2^.

### 3.2. Experiencing Food Restrictions in Childhood and Early Adulthood 

More women than men restricted some foods, and more women limited eating processed food, sugar, and fat-rich foods compared to men. However, no differences were found between women and men in restricting sugar-rich foods (Table 2). Twice as many men (33.5%) as women (16.5%) did not apply any restrictions, while almost 4 times as many women (24.7%) as men (6.3%) limited their consumption of all types of food, i.e., processed foods, sugar, sugar-rich foods, and fat-rich foods. Differences between respondents with normal body weight and those with overweight were found only for sugar-rich foods. More people with normal body weight limited their intake of sugar-rich foods compared to those with overweight (Table 2).

Descriptive statistics for restrained eating and food experiences from childhood related to restrictions are presented in Table 3.

Childhood food experiences related to restrictions for health and weight control were positively interrelated (r = 0.545; *p* < 0.01) (Table 4). A weak positive correlation was found between childhood experiences of weight control restrictions and restrained eating in early adulthood (r = 0.222; *p* < 0.01). Childhood experiences of restrictions for health did not correlate with either restrained eating or food restrictions in early adulthood. However, experiences of restrictions on weight control from childhood showed a weak positive correlation with using restrictions in general, the number of restrictions, and restrictions on sugar and food rich in fat. Restrained eating in early adulthood was correlated with restricted consumption of highly processed foods (0.302), sugar (0.384), high-sugar foods (0.414), and high-fat foods (0.440) (Table 4).

### 3.3. Predictors of Having Excessive Weight

The likelihood of having excessive body weight increased with an increase in having food experiences related to weight control restrictions by 57% (crude model) and by 53% in the model adjusted for sex and age. Experiences of restrictions for health, restrictions on eating foods, and the number of applied restrictions were not predictors of having excessive body weight (Table 5).

## 4. Discussion

This study focused on understanding how restrained eating in early adulthood, food-restricted types, and food experiences related to restrictions originating in childhood are linked to excessive body weight in young adults. We found that childhood food experiences of weight control restrictions were positively associated with having excessive weight in early adulthood. On the other hand, the relationships between excessive body weight and such variables as restrained eating, food-restricted types, and childhood food experiences related to health restrictions were not confirmed.

The relationship between childhood food experiences related to weight control, restrictions, and excessive body weight in young adulthood is in line with results suggesting that excessive body weight is determined by parental dietary control [39]. Parents’ perceptions of their child’s current and future weight status influence their feeding practices, leading to greater control and monitoring of their child’s diet [40]. Differences in the mother’s perception of the child’s health can lead to overestimating and underestimating the child’s actual weight, resulting in inappropriate feeding practices such as dietary restrictions or excessive eating pressure, potentially affecting future body mass index [40,41,42]. Meanwhile, the experience of weight control can be a robust early stressor for some children, which, like other adverse childhood experiences, can result in overweight or obesity [37,43]. It is emphasized that motivating children to lose weight and criticizing their weight can lead to decreased physical self-esteem and increased risk of maladaptive eating behaviors, such as dieting and eating disorders [44]. Moreover, food restrictions predict loneliness among children and adults [45]. 

The findings have shown positive correlations between restrained eating in young adults and the application of some food restrictions, particularly concerning consuming processed foods, sugar, and products high in fats and sugar. This indicates that individuals who score higher on restrained eating are more likely to make conscious decisions to exclude or limit unhealthy foods, which may be associated with their commitment to maintaining a healthy lifestyle. Previous studies indicate that individuals who implement dietary restrictions are more inclined to follow healthier patterns rich in fruits and vegetables, limiting their consumption of processed foods and high-calorie snacks [46]. Despite the more health-beneficial choices accompanying food restrictions, the potential risks of overly restrictive diets that can lead to nutrient deficiencies must also be considered [47]. 

Both restrained eating and food-restricted types were not predictors of excessive body weight. This may be because restrained eating expresses attempts at dietary restriction [48]. It is still unclear how often restrained eating results in effective dietary restriction [49]. In addition, dietary restrictions encompass a wide spectrum of restrictions introduced for various reasons. Medical reasons such as allergies, intolerances, or certain health conditions may determine dietary restrictions more than body weight [26]. The evidence for restrained eating and BMI is conflicting, but many studies show that restrained eating scores are positively associated with BMI [50,51,52,53]. Although restrained eating may predict weight gain, having a high BMI may also favor high restraint scores. In a group of people with obesity, the desire to lose weight, which favors food restriction, increases with increasing body weight [54]. Nonetheless, previous studies indicate that restrained eating is not related to BMI [55], suggesting unsuccessful attempts to avoid certain foods or limit calorie intake [49]. According to goal conflict theory, weight regulation problems arise from the conflict between the goal of weight control and eating pleasure, which should be considered when explaining the outcome discrepancy [56]. The weight control goal is often subordinated to hedonistic expectations about food [57]. High dietary restrictions have been associated with heightened reactivity towards food [58]. In this situation, attempts to restrict eating may increase food cravings and intrusive thoughts about the desired food, which increases the risk of further overeating behaviors and weight gain [59]. 

When analyzing gender differences, women were more likely to restrict all categories of food included in the study simultaneously. This confirmed that women are more inclined to control their diets rigorously [60,61,62]. Women also exhibit higher dietary restriction levels than men, limiting the intake of processed foods, sugar, and fat [61]. On the other hand, men more frequently report no dietary restrictions and are more likely to choose high-fat foods (such as red meat, meat products, and eggs) and eat meals out more often [63]. This suggests that their approach to nutrition is more oriented toward the pleasure of eating [60]. Only differences in the use of restrictions in sugar-rich foods were shown between young adults with normal and excessive body weight. In contrast, those groups had no differences in limiting processed, sugar, and fat-rich foods. These results are partly consistent with what is known from earlier studies. A higher BMI was conducive to regulating the healthiness of meals but lowered the tendency to regulate portion size or calorie content [64], which may explain the lack of difference in limiting processed foods, sugar, and fat-rich foods. The consumption of more red meat, processed meat, and cheese by overweight adolescents or adolescents with obesity than by healthy-weight adolescents [65,66], in the absence of differences in limiting intake of high-energy foods, may indicate further weight gain.

Based on the link between childhood experiences and restricting high-calorie foods in adulthood, it is reasonable to speculate that parental dietary practices aimed at restricting foods to control a child’s weight may be more critical in shaping adult behavior related to high-calorie foods than practices aimed at restricting foods to ensure health. However, having experienced restrictions for weight control originating in childhood may contribute to higher body weight in adulthood. Many studies, including longitudinal ones, have already indicated an adverse effect of a high restriction of sugar-rich foods and beverages [67]. Controlling practices such as the restriction of sugar-rich foods and beverages, the use of sugar-rich food to soothe, and rewards exhibit long-term potential for developing negative behaviors in children and adolescents. Therefore, parents should refrain from using such approaches to deal with sugar-rich foods to promote the development of self-regulatory competencies in children [68]. 

### 4.1. The Implications of the Study

The results suggest that parents may unwittingly promote excessive weight gain in early adulthood by using inappropriate eating behaviors in children. Therefore, interventions should be developed to increase parents’ awareness of the possible long-term consequences of eating behavior control practices in children. Thus, the knowledge generated by this study can be used by educators who communicate with parents of young children to create educational messages. Informing parents about the possible consequences of controlling their child’s weight and drawing their attention to two types of dietary restrictions, i.e., weight control restrictions and health restrictions, is crucial from the perspective of counteracting excessive weight in childhood and adulthood.

### 4.2. The Strengths and Limitations 

Although this is a cross-sectional study in which young adults report on their childhood food experiences, the inclusion of retrospective questions about childhood food experiences made it possible to estimate their relationship with restricting practices in young adulthood. Limitations of the study include using a convenience sample, which limits the possibility of generalizing the results to Polish young adults and other young populations due to differences related to ethnic origin or socio-economic status. Another limitation is the lack of information on ethnicity and socio-economic status. The results are derived from measuring restrained eating, childhood eating experiences, dietary restrictions, and anthropometric variables as self-reports. Self-reporting may not be sufficient to classify individuals as having excessive weight. Other variables may also be biased for this reason. Moreover, childhood eating experiences were based on retrospective reports, which may be inaccurate due to memory constraints. However, even if participants’ memories of eating were inaccurate, remembered childhood experiences of weight control may underlie current daily activities. To address the limitation of the study’s retrospective nature, it is suggested that longitudinal cohort studies tracking the impact of parental feeding practices from birth to adulthood be conducted. This would provide greater insight into the role of parental feeding practices related to child weight control in shaping restrictive behavior in later life. Such a study would also provide some insight into other factors that may modify the effect of parental practices. The causality of the association between variables cannot be determined from the results of our study due to its cross-sectional nature, nor can we account for the importance of different factors influencing restrictive behavior between childhood and early adulthood. 

### 4.3. The Direction for Future Research

The psychological consequences of weight control in childhood may manifest in adulthood as the experience of negative emotions related to eating and difficulties in controlling one’s behavior. This may facilitate the use of dietary restraint. Further studies focusing on weight control through restrictive practices should include psychological tests and assessing eating behaviors to confirm the relationship between the psychological functioning of individuals experiencing restraint in childhood and eating behaviors in adulthood. Mixed results showing that restrained eating is associated with lower body weight and susceptibility to obesity and overeating [69] or the lack of such an association [55], as in our study, indicate that further research on restrictive eating and excess body weight is warranted. Critical in these studies is the inclusion of variables differentiating the groups (e.g., reasons for restraint) and different tools for assessing restrained eating [26,70]. Because educating parents about parenting practices other than restrictive weight control measures, such as monitoring or modeling, seems promising [71], we suggest that future research expand the scope of parenting practices beyond those related to food restrictions. Due to the limitations of cross-sectional studies, it is recommended that longitudinal studies be conducted to assess the cause-and-effect relationship between childhood food experiences, restrained eating in young adulthood, and body weight status. However, currently, there is scant evidence to inform guidance for parents and caregivers regarding alternatives to the use of restriction. Future research is needed to identify alternative feeding strategies that successfully promote moderation in children’s intake of preferred, palatable foods and foster the development of self-regulation in children. Moreover, future research should explore how other eating styles, in addition to the restrained eating and childhood eating experiences from restrictive practices, may influence dietary restrictions in young people. This would provide an opportunity for a more comprehensive approach in preparing recommendations for parents on managing their child’s feeding to prevent excessive weight gain in childhood and adulthood. 

## 5. Conclusions

This study has shown that experiencing “Restrictions for weight control” in childhood correlates with adults’ restrained eating and food restrictions in young adulthood. In the study group, restrained eating in adulthood and current food restrictions were not predictors of having excessive body weight. Only childhood food experiences of restrictions for weight control increased the likelihood of having excessive body weight in the sample. The results suggest that further research on the relationship between childhood experiences of restrictions for weight control and excessive body weight in adulthood is needed to confirm these findings. Their outcomes could confirm the need to develop interventions to increase parents’ awareness of the possible long-term consequences of practices related to controlling children’s eating behavior.

## Figures and Tables

**Table 1 nutrients-17-00087-t001:** Characteristics of the study sample (N = 358).

Socio-Demographic Characteristics	Total Sample % (N)
Sex	
Female	50.8 (182)
Male	49.2 (176)
Education	
Lower than secondary	13.7 (49)
Secondary	66.7 (239)
Higher	19.6 (70)
Place of residence	
A village	19.6 (70)
A town with less than 50,000 inhabitants	17.9 (64)
A town with 50–250 inhabitants	10.9 (39)
A city with over 250,000 inhabitants	51.6 (185)
BMI	
18.5–24.99 kg/m^2^	76.3 (273)
25 kg/m^2^ and above	23.7 (85)
Age (in years—median, range)	21.0 (18–25)
BMI (in kg/m^2^—median, range)	22.6 (18.5–45.7)

N-number of participants.

**Table 2 nutrients-17-00087-t002:** Restrictions on eating food in early adulthood by sex and BMI.

Restrictions			Sex		BMI	
		Total % (N)	Female % (N)	Male % (N)	Normal Weight % (N)	Excessive Weight % (N)
		N = 358	N = 182	N = 176	N = 273	N = 85
Restrictions on eating foods		75.1 (269)	83.5 ^c^ (152)	66.5 ^c^ (117)	74.4 (203)	77.6 (66)
Restrictions on eating processed foods		37.2 (133)	48.4 ^c^ (88)	25.6 ^c^ (45)	38.1 (104)	34.1 (29)
Restrictions on eating sugar		59.2 (212)	64.8 ^a^ (118)	53.4 ^a^ (94)	57.5 (157)	64.7 (55)
Restrictions on eating sugar-rich foods		46.1 (165)	46.9 (128)	43.5 (37)	52.7 ^b^ (96)	39.2 ^b^ (69)
Restrictions on eating fat-rich foods		37.7 (135)	53.3 ^a^ (97)	21.6 ^a^ (38)	38.5 (105)	35.3 (30)
Number of restrictions applied	None	24.9 (89)	16.5 ^c^ (30)	33.5 ^c^ (59)	25.6 (70)	22.4 (19)
	1 type of restriction	18.2 (65)	14.3 ^c^ (26)	22.2 ^c^ (39)	17.6 (48)	20.0 (17)
	2 types of restriction	24.6 (88)	27.5 ^c^ (50)	21.6 ^c^ (38)	23.8 (65)	27.1 (23)
	3 types of restriction	16.8 (60)	17.0 ^c^ (31)	16.5 ^c^ (29)	16.2 (44)	18.7 (16)
	4 types of restriction	15.6 (60)	24.7 ^c^ (45)	6.3 ^c^ (11)	16.8 (46)	11.8 (10)

N—number of respondents; ^a^
*p* < 0.05; ^b^
*p* < 0.01; ^c^
*p* < 0.001 (chi-squared test).

**Table 3 nutrients-17-00087-t003:** Restrained eating in early adulthood and food experiences from childhood related to restrictions—descriptive statistics.

Variables			Statistics
Restrained eating in early adulthood	Mean		2.67
	95% Confidence Interval for Mean	Lower Bound	2.60
		Higher Bound	2.74
	Median		2.60
	Variance		0.89
	Std. deviation		0.94
	Minimum		1.00
	Maximum		5.00
	Skewness		0.24
	Kurtosis		−0.70
Experiences of restrictions for health	Mean		2.82
	95% Confidence Interval for Mean	Lower Bound	2.71
		Higher Bound	2.92
	Median		2.75
	Variance		1.02
	Std. Deviation		1.01
	Minimum		1.00
	Maximum		5.00
	Skewness		0.02
	Kurtosis		−0.72
Experiences of restrictions for weight control	Mean		2.82
	95% Confidence Interval for Mean	Lower Bound	2.71
		Higher Bound	2.92
	Median		1.75
	Variance		0.68
	Std. deviation		0.82
	Minimum		1.00
	Maximum		4.75
	Skewness		1.02
	Kurtosis		0.73

**Table 4 nutrients-17-00087-t004:** Bivariate correlations between restrictions on eating and restrained eating in early adulthood and food experiences of restrictions for health and weight control from childhood (Spearman’s correlation coefficient).

Variables	Experiences of Restrictions for Health	Experiences of Restrictions for Weight Control	Restrained Eating
Experiences of restrictions for health	1	0.545 **	0.087
Experiences of restrictions for weight control	0.545 **	1	0.222 *
Restrictions on eating foods	0.030	0.143 **	0.481 **
Restrictions on eating processed foods	0.003	0.088	0.302 **
Restrictions on eating sugar	−0.006	0.149 **	0.384 **
Restrictions on eating sugar-rich foods	−0.083	0.104	0.414 **
Restrictions on eating fat-rich foods	−0.006	0.105 *	0.440 **
Number of restrictions applied	0.029	0.152 **	0.545 **

* *p* < 0.05; ** *p* < 0.01.

**Table 5 nutrients-17-00087-t005:** Odds ratios for excessive weight according to restrictions on eating food, restrained eating in early adulthood and food experiences of restrictions for health and weight control from childhood.

Variables	Model	β *	e^β^ *	95CI		*p*-Value **
Restrained eating in adulthood	Crude	0.009	1.009	0.977	1.042	0.593
	Adjusted *	0.032	1.032	0.996	1.069	0.078
Childhood experiences of restrictions for weight control	Crude	0.453	1.573	1.104	2.242	0.012
	Adjusted	0.428	1.535	1.069	2.204	0.020
Childhood experiences of restrictions for health	Crude	0.075	1.078	0.801	1.451	0.622
	Adjusted	0.052	1.054	0.775	1.432	0.738
Restrictions on eating foods in adulthood	Crude	0.379	1.461	0.620	3.445	0.386
	Adjusted	0.268	1.307	0.541	3.156	0.552
Number of restrictions applied in adulthood	Crude	−0.188	0.828	0.626	1.097	0.188
	Adjusted	−0.134	0.874	0.654	1.168	0.363
Constant	Crude	−2.487	0.083			<0.001
	Adjusted	−5.727	0.003			<0.001

Adjusted for sex and age *, OR—point estimate (β e^β^), 95% confidence intervals; ** significance level of the Wald’s test.

## Data Availability

The data are not publicly available because they have not yet been made available in ‘publicly available databases’. However, the data presented in the study are available upon request from the corresponding author.

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
