# Peer review of "Experiencing Food Restrictions for Health and Weight Control in Childhood and Their Links to Restrained Eating and Excessive Body Weight in Polish Young Adults—A Cross-Sectional Study"

_nutrients, 2024, doi:10.3390/nu17010087_

Round 1
Reviewer 1 Report (New Reviewer)
Comments and Suggestions for Authors
Cross sectional survey amongst 358 young Polish adults relating current restrained eating to recalled parental dietary restrictions during their childhood . The authors conclude that diet restriction framed to prevent weight in childhood is associated with higher weight in young adults and this requires further work on best ways for parents to manage children’s feeding to prevent weight problems.
This has some interest but would benefit from being a shorter more focussed report. There are a number of key limitations the authors need to acknowledge , which should modify their definitive conclusions.
Introduction
Needs be shortened and cite the extent and reasons for weight problems and the potential role of childhood eating patterns and parental influence. Given that weight problems are largely driven by the obesogenic environment and access to unhealthy high fat / sugar foods, clearly some restriction is required or they would have unlimited calories and weight problems. The issue is about the framing of the restriction and how this is best done
Method
· Need more info on social media recruitment. Where was it posted? & how quickly was the cohort recruited ? Were there any reimbursements &
· Need to inform reader which questions in the questionnaire were scored for the restriction and weight loss and restriction for health sub scales
· Need to justify sample size. Study was powered 207 medium effect size chi squared for differences in early adulthood restraints and BMI and gender and does not seem to be powered for the regression undertaken which is the main research question of the study. Need justification for undertaking regression on the questionnaire responses which are continuous data with just 5 different values .
Results
· Table 1 requires units for all variables
· Need to say how many (%) were obese ?
· Table 3 not need all stats as a 5 scale point can just report the mean ( SD)
· Table 5 need to be clear in variable list that restrained eating is during adulthood
Discussion
Need to acknowledge key limitations
· Need to emphasise the study is just flagging association not necessarily causal .
· Should have asked participants about their childhood weight status, and that of their parents:
· It is quite possible that children who had food restriction during childhood could have already have weight issues which then tracked through to adulthood. This does not mean the adult weight problem was caused by the parental dietary restrictions. The association may be bidirectional with more restrained eating advised if the child was heavier.
· Similarly, children of parents with weight problems / disordered eating may have been more restricted and may be heavier because of a genetic predisposition to weight problems rather than that they were restricted during childhood.
· Would have been interesting to look at relationship between childhood restriction and current weight in men & women
· Also, prevalence of any current or previous eating disorders
· Need to say how much participants were told of the aims of the study as the patient’s information sheet for recruitment has the potential to attract a biased sample with disordered eating
· Another limitation is lack of information on ethnicity and socio economic status which won’t necessarily be captured by education or are of residence
Other discussion points
Discussion needs to be greatly shortened
i.e. Omit line 238 -243 as off point
Lose this line
“The results of our study indicate that gender plays a more significant role than body 274 mass index (BMI) in differentiating behaviours related to food restriction. “and just say more restrained in women in the survey
lose the discussion on different adult food restraints bit vague & does not add lines 282-291
The authors should acknowledge the large body of literature on evidence based approaches to promote healthy eating / weight control in children which the authors have not cited and how this is managed in research and clinical practice.
Minor point
Assume this should say non – nutritious foods
However, limited access to inexpensive and nutritious food (e.g., supermarkets), 49 referred to as a ‘food desert’ [11], is also positively associated with obesity
Author Response
Comment 1. Cross sectional survey amongst 358 young Polish adults relating current restrained eating to recalled parental dietary restrictions during their childhood . The authors conclude that diet restriction framed to prevent weight in childhood is associated with higher weight in young adults and this requires further work on best ways for parents to manage children’s feeding to prevent weight problems. This has some interest but would benefit from being a shorter more focused report. There are a number of key limitations the authors need to acknowledge , which should modify their definitive conclusions.
Response to comment 1. Thank you very much for these comments. We kindly thank you for the time and effort taken to read and review our article.
Comment 2. Introduction. Needs be shortened and cite the extent and reasons for weight problems and the potential role of childhood eating patterns and parental influence. Given that weight problems are largely driven by the obesogenic environment and access to unhealthy high fat / sugar foods, clearly some restriction is required or they would have unlimited calories and weight problems. The issue is about the framing of the restriction and how this is best done
Response to comment 2. The comments were taken into account when improving the text. The Introduction was shortened.
Comment 3. Need more info on social media recruitment. Where was it posted? & how quickly was the cohort recruited ? Were there any reimbursements?
Response to comment 3. The invitations were posted on Facebook groups of students participating in the class Food Sociology at SGGW. Next to the request to fill out the questionnaire, there was also a request to share the link in other groups where the person was active. The cohort was recruited within 1 month. There were no problems with the implementation of the study.
Comment 4. Need to inform reader which questions in the questionnaire were scored for the restriction and weight loss and restriction for health sub scales.
Response to comment 4. This information was added to the questionnaire in the Supplementary.
Comment 5. Need to justify sample size. Study was powered 207 medium effect size chi squared for differences in early adulthood restraints and BMI and gender and does not seem to be powered for the regression undertaken which is the main research question of the study. Need justification for undertaking regression on the questionnaire responses which are continuous data with just 5 different values .
Response to comment 5. Regression was performed for continuous data ranging from 1 to 5. However, these were not just 5 different values because the mean values were calculated based on the subscales. The sum of ratings was divided by the number of items, which gives many different values ranging from 1 to 5, which justifies treating this variable as a quantitative variable.
Comment 6. Table 1 requires units for all variables
Response to comment 6. According to the suggestion, units were added.
Comment 7. Need to say how many (%) were obese ?
Response to comment 7. Only 5.3% of respondents were obese, so we decided to identify the group with excessive body weight instead of two separate groups (overweight and obese). This is also justified for the research aim.
Comment 8. Table 3 not need all stats as a 5 scale point can just report the mean ( SD)
Response to comment 8. In our opinion, the remaining indicators can also be presented on a 5-point scale.
Comment 9. Table 5 need to be clear in variable list that restrained eating is during adulthood
Response to comment 9. Such a clarification was made.
Comment 10. Discussion. Need to acknowledge key limitations
Response to comment 10. The key limitation were acknowledged.
Comment 11. Need to emphasise the study is just flagging association not necessarily causal.
Response to comment 11. It was emphasized that this study is only intended to demonstrate a relationship, not a cause-and-effect relationship.
Comment 12. Should have asked participants about their childhood weight status, and that of their parents:
- It is quite possible that children who had food restriction during childhood could have already have weight issues which then tracked through to adulthood. This does not mean the adult weight problem was caused by the parental dietary restrictions. The association may be bidirectional with more restrained eating advised if the child was heavier.
- Similarly, children of parents with weight problems / disordered eating may have been more restricted and may be heavier because of a genetic predisposition to weight problems rather than that they were restricted during childhood.
Response to comment 12. We agree with this opinion. Nevertheless, it is challenging to expect respondents to remember their childhood weight. We could only ask if they were too big. Unfortunately, we did not ask about this.
Comment 13. Would have been interesting to look at relationship between childhood restriction and current weight in men & women
Response to comment 13. Such analyses were performed. There were no differences in women when weight (BMI) was considered. In men, there was a difference between men with normal and excessive body weight. Those with a BMI above 25 kg/m2 had higher scores on weight control experiences (2.24) than those with a BMI below 25 kg/m2 (1.78).
Comment 14. Also, prevalence of any current or previous eating disorders.
Response to comment 14. The eating disorders were not measured.
Comment 15. Need to say how much participants were told of the aims of the study as the patient’s information sheet for recruitment has the potential to attract a biased sample with disordered eating
Response to comment 15. The aim of the study was formulated in a very general way. In the invitation to the survey, information was presented that the objective of the study was to learn about the relationship between childhood food experiences and current eating styles. Therefore, such a formulation of the aim should not impact accepting the invitation to the study.
Comment 15. Another limitation is lack of information on ethnicity and socio economic status which won’t necessarily be captured by education or are of residence
Answer to comment 15. We are aware of this limitation. It has been added to the study limitations.
Comment 16. Discussion needs to be greatly shortened. i.e. Omit line 238 -243 as off point. Lose this line “The results of our study indicate that gender plays a more significant role than body 274 mass index (BMI) in differentiating behaviours related to food restriction. “and just say more restrained in women in the survey. Lose the discussion on different adult food restraints bit vague & does not add lines 282-291
Answer to comment 16. We have removed this fragment, although it was introduced at the request of another reviewer. This shows how differently we look at texts prepared by other authors.
Comment 17. The authors should acknowledge the large body of literature on evidence based approaches to promote healthy eating / weight control in children which the authors have not cited and how this is managed in research and clinical practice.
Answer to comment 17. Thank you for this comment. Our attention was focused on restrictions, hence the topic of promoting healthy eating was not taken up.
Comment 18. Assume this should say non – nutritious foods
However, limited access to inexpensive and nutritious food (e.g., supermarkets), referred to as a ‘food desert’ [11], is also positively associated with obesity
Answer to comment 18. The term “nutritious food” was eliminated.
Reviewer 2 Report (New Reviewer)
Comments and Suggestions for Authors
Childhood obesity is relevant to the global health crisis. Children with overweight or obesity may develop obesity in adulthood. Eating behaviour training is important for the prevention of obesity in both childhood and adulthood. This study is thus relevant. Please note that eating behaviour training is very different from food restriction in childhood, which may be resulted from regional famine and cause direct death or later life health issues (doi: 10.2139/ssrn.4848499). Please revise the title and state clearly its difference from causing young people malnutrition.
1. The major limitation of this study is the recall bias. It is thus biased to exclude people without response. Instead, sensitivity analyses by assuming the both direction are necessary to show the robustness of this study.
2. Another limitation is regarding the question, which can be vague for responders. The answer is likely arbitary with subjective judgement of food restriction. It warrants strict statement in the Discussion section.
3. Please note that body composition including the body fat mass and lean mass are relevant to later lives (doi: 10.1111/dom.15373). Please indicate the relevance of the eating behaviour training with the body composition with and without the study findings.
4. Please remind that eating behaviour training in childhood is not only linked to later life obesity but also other health outcomes. It is necessary to consider other outcomes or comorbidities in this study. If infeasible, please discuss carefully.
Comments on the Quality of English LanguageThe manuscript used many paraphrases. The food restriction in the title is very confusing.
Author Response
Comment 1. Childhood obesity is relevant to the global health crisis. Children with overweight or obesity may develop obesity in adulthood. Eating behaviour training is important for the prevention of obesity in both childhood and adulthood. This study is thus relevant. Please note that eating behaviour training is very different from food restriction in childhood, which may be resulted from regional famine and cause direct death or later life health issues (doi: 10.2139/ssrn.4848499). Please revise the title and state clearly its difference from causing young people malnutrition.
Response to comment 1. Thank you for this comment. We have changed the title to "Experience of dietary restrictions for health and weight control in childhood and their associations with restrained eating and excessive body weight in young adults in Poland - a cross-sectional study". We hope that now it is evident that the restrictions we studied should not be associated with malnutrition.
Comment 2. The major limitation of this study is the recall bias. It is thus biased to exclude people without response. Instead, sensitivity analyses by assuming the both direction are necessary to show the robustness of this study.
Response to comment 2. We are aware that the limitation of our study is due to recall errors. Some individuals were excluded from the sample due to a lack of information on childhood eating experiences. ”From the group of respondents that accepted the invitation and adequately filled the questionnaire, 77 respondents were excluded due to their inability to remember food experiences from childhood related to restrictions, i.e., those who answered “I don’t remember” for any question or statement related to food experiences.“ Therefore, the exclusion took place when the study group was created. Hence, the analyses were performed without missing values.
Comment 3. Another limitation is regarding the question, which can be vague for responders. The answer is likely arbitary with subjective judgement of food restriction. It warrants strict statement in the Discussion section.
Response to comment 3. The specificity of this study, like other questionnaire studies, is the collection of subjective answers, which are then analyzed and interpreted.
Comment 4. Please note that body composition including the body fat mass and lean mass are relevant to later lives (doi: 10.1111/dom.15373). Please indicate the relevance of the eating behaviour training with the body composition with and without the study findings.
Response to comment 4. We did not study body composition, so this comment is difficult to include in our work.
Comment 5. Please remind that eating behaviour training in childhood is not only linked to later life obesity but also other health outcomes. It is necessary to consider other outcomes or comorbidities in this study. If infeasible, please discuss carefully.
Response to comment 5. Discussing other outcomes or comorbidities in this study is not feasible due to the lack of such data. It is not entirely clear why, given the impossibility of this request, we should discuss it in detail.
Round 2
Reviewer 2 Report (New Reviewer)
Comments and Suggestions for Authors
Thanks for the revision and I have no more comments.
This manuscript is a resubmission of an earlier submission. The following is a list of the peer review reports and author responses from that submission.
Round 1
Reviewer 1 Report
Comments and Suggestions for Authors
The manuscript by Jeżewska-Zychowicz et al. is well-written and investigates the effects of restricted eating and childhood food experiences on young adults' excessive body weight. They found a positive association between experiencing weight control restrictions in childhood and excessive body weight in early adulthood.
The authors conclude with advising to develop interventions to increase parents’ awareness of the potential long-term outcomes of the practices related to eating and food during childhood.
Minor points:
- Line 44, “obese children” (used twice) should be changed to “children with obesity”.
- Line 319, “obese adolescents” should be changed to person-first language “adolescents with obesity”.
- It would benefit the manuscript if the authors could speak to the epigenetic changes of periods of famine/food inaccessibility.
Reviewer 2 Report
Comments and Suggestions for Authors
The study by Jeżewska-Zychowicz et al., "Experiencing Food Restrictions in Childhood and Early Adulthood as Predictors of Excessive Body Weight in Polish Young Adults," explores the impact of childhood food restriction experiences on excessive body weight in Polish adults aged 18-25. Through a cross-sectional survey with 358 participants, the authors examined both past and current food restriction practices.
Findings reveal that childhood weight-control restrictions are significantly linked to excessive body weight in young adulthood, independent of health-oriented restrictions or the type and number of foods restricted. Although these early restrictions correlated weakly with restrained eating habits and the limitation of processed, sugary, and fat-rich foods in adulthood, there was no direct association between current food restrictions and excessive weight. This suggests that weight-control restrictions in childhood may have a more lasting influence on weight outcomes than dietary behaviors in adulthood.
· This study is exceptionally well-written and executed—congratulations.
· It addresses an important yet often overlooked public health issue.
· The findings provide valuable guidance for designing parental interventions to reduce the risk of excessive weight gain in young people.
· Limitations of the study design have been comprehensively addressed.
Only one comment: could you please add the questionnaire(s) used in the study as supplemental? This would be useful.
Reviewer 3 Report
Comments and Suggestions for Authors
1) The design of the study needs to be clearly mentioned in the title
2) Due to cross-sectional design of the study, any statement regarding the cause-effect relationship between obesity and dietary behaviours should be softened and rounded, in all the sections of this manuscript from the abstract to the conclusion
3) The keywords should be different from those, which appear in the title
4) The introduction section is too long, should be more focused and less referenced, by the way kindly avoid self-citation when not necessary
5) Do the authors used the validated polish version of the questionnaires, if yes, please indicate the reference of validation
6) How was the sample determined? A power analysis has been conducted? If yes, this should be included.
7) Authors reported that the data were collected in Poland in 2020-2021, but the approved by the Ethics Committee of the Warsaw University of Life 384 Sciences in Poland (Resolution No. 8/RKE/2023/U, 20 April 2023), this is problematic while conducting research
8) The discussion section should be better organized as following:
· Main finding of the study and comparison with previous published finding
· The implication of the study
· The strengths and limitations
· The direction for future research